# IMPROVING DEEP REGRESSION WITH TIGHTNESS

**Shihao Zhang**[1], **Yuguang Yan**[2], **Angela Yao**[1]
[1]National University of Singapore [2]Guangdong University of Technology
zhang.shihao@u.nus.edu    ygyan@gdut.edu.cn    ayao@comp.nus.edu.sg

## ABSTRACT

For deep regression, preserving the ordinality of the targets with respect to the feature representation improves performance across various tasks. However, a theoretical explanation for the benefits of ordinality is still lacking. This work reveals that preserving ordinality reduces the conditional entropy $\mathcal{H}(\mathbf{Z}|\mathbf{Y})$ of representation $\mathbf{Z}$ conditional on the target $\mathbf{Y}$. However, our findings reveal that typical regression losses fail to sufficiently reduce $\mathcal{H}(\mathbf{Z}|\mathbf{Y})$, despite its crucial role in generalization performance. With this motivation, we introduce an optimal transport-based regularizer to preserve the similarity relationships of targets in the feature space to reduce $\mathcal{H}(\mathbf{Z}|\mathbf{Y})$. Additionally, we introduce a simple yet efficient strategy of duplicating the regressor targets, also with the aim of reducing $\mathcal{H}(\mathbf{Z}|\mathbf{Y})$. Experiments on three real-world regression tasks verify the effectiveness of our strategies to improve deep regression. Code: https://github.com/needylove/Regression_tightness.

## 1 INTRODUCTION

Classification and regression are two fundamental tasks in machine learning. Classification maps input data to categorical targets, while regression maps the data to continuous target space. Representation learning in deep neural networks is well-studied for classification (Boudiaf et al., 2020; Achille & Soatto, 2018), but is less explored for regression. One emerging observation in deep regression is the importance of feature ordinality (Zhang et al., 2023). Preserving the ordinality of targets within the feature space leads to better performance, and various regularizers to enhance ordinality have been proposed (Gong et al., 2022; Keramati et al., 2023). But what is the underlying link between ordinality and regression performance?

The information bottleneck principle (Shwartz-Ziv & Tishby, 2017) suggests that a neural network learns representations $\mathbf{Z}$ that retain sufficient information about the target $\mathbf{Y}$ while compressing irrelevant information. The two aims can be regarded as minimizing the conditional entropies $\mathcal{H}(\mathbf{Y}|\mathbf{Z})$ and $\mathcal{H}(\mathbf{Z}|\mathbf{Y})$ (Zhang et al., 2024). Compression reduces representation complexity, prevents overfitting, and bounds the generalization error (Tishby & Zaslavsky, 2015; Kawaguchi et al., 2023; Zhang et al., 2024). We find that preserving ordinality enhances compression by minimizing $\mathcal{H}(\mathbf{Z}|\mathbf{Y})$, *i.e.*, the conditional entropy of the learned representation $\mathbf{Z}$ with respect to the target $\mathbf{Y}$. Following (Boudiaf et al., 2020; Zhang et al., 2024), we refer to this conditional entropy as *tightness*, and its compression as *tightening the representation*.

But are ordinal feature spaces not learned naturally by the regressor? We explore this question through gradient analysis and comparing the differences between regression and classification. We find that typical regressors are weak in tightening the learned representations. Specifically, given a fixed linear regressor with weight vector $\boldsymbol{\theta}$, the update direction of $\mathbf{z}_i$ for a given sample $i$ tends to follow the direction of $\boldsymbol{\theta}$. The movement of $\mathbf{z}_i$ can be regarded as a probability density shift (Sonoda & Murata, 2019). Deep regressors update and tighten the representation in limited directions perpendicular to $\boldsymbol{\theta}$. In contrast, we find that deep classifiers update $\mathbf{z}_i$ more flexibly and in diverse directions, leading to better-tightened representations. Such a finding sheds insight into why reformulating regression as a classification task may be more effective (Farebrother et al., 2024; Liu et al., 2019) and why classification losses benefit representation learning for regression (Zhang et al., 2023).

So how can regression representations be further tightened? We take inspiration from classification, where one-hot encodings allow a separate set of classification weights $\boldsymbol{\theta}_k$ for each class $k$. Similarly,

we augment the target space of the regressor into multiple targets. The multiple-target strategy adds extra dimensions to the regression output and incorporates additional regressors, making it more flexible to tighten the feature representations. Additionally, we introduce a Regression Optimal Transport (ROT) Regularizer, or ROT-Reg. ROT-Reg captures local similarity relationships through optimal transport plans. By encouraging similar transport plans between the target and feature space, the regularizer can tighten representations locally. It also helps to preserve the target space topology, which is also desirable for regression representations (Zhang et al., 2024).

Our main contributions are three-fold:

- We are the first to analyze the need for preserving target ordinality with respect to the representation space for deep regression and link it to feature tightness.

- We reveal the weakness of standard regression in tightening learned feature representations, as the representation updating direction is constrained to follow a single line.

- We introduce a multi-target learning strategy and an optimal transport-based regularize, which tighten regression representations globally and locally, respectively.

## 2  RELATED WORK

**Regression representation learning**. Existing works mainly focus on the properties of continuity and ordinality. For continuity, DIR (Yang et al., 2021) tackles missing data by smoothing, based on the continuity of both targets and representations. VIR (Wang & Wang, 2024) computes representations with additional information from data with similar targets. Preserving representation continuity also encourages the feature manifold to be homeomorphic with respect to the target space and is highly desirable (Zhang et al., 2024).

For ordinality, RankSim (Gong et al., 2022) explicitly preserves ordinality for better performance. (Keramati et al., 2023) further incorporated a contrastive regularizer to preserve ordinality. It is worth mentioning that continuity overlaps with ordinality conceptually; ensuring continuity for neighboring samples inherently involves ordinality. Although ordinality plays a key role in regression representation learning, its importance and characteristics are underexplored. This work tackles these questions by establishing connections between target ordinality and representation tightness.

**Recasting regression as classification**. For a diverse set of regression tasks, formulating them into a classification problem yields better performance (Li et al., 2022; Bhat et al., 2021; Farebrother et al., 2024). Previous works have hinted at task-specific reasons. For pose estimation, classification provides denser and more effective supervision (Gu et al., 2022). For crowd counting, classification is more robust to noise (Xiong & Yao, 2022). Later, Pintea et al. (2023) empirically found that classification helps when the data is imbalanced, and Zhang et al. (2023) suggests that regression lags in its ability to learn a high entropy feature space. A high entropy feature space implies that representations preserve necessary information about the target. In this work, we provide a derivation and further suggest that regression struggles to compress the representations.

## 3  ON THE TIGHTNESS OF REGRESSION REPRESENTATIONS

### 3.1  NOTATIONS & DEFINITIONS

Consider a dataset $\{\mathbf{x}_i, \mathbf{z}_i, y_i\}_{i=1}^N$ sampled from a distribution $\mathcal{P}$, where $\mathbf{x}_i$ is the input, $y \in \mathbb{R}$ is the corresponding label, and $\mathbf{z}_i \in \mathcal{Z} \subset \mathbb{R}^d$ is the feature corresponding to the input $\mathbf{x}_i$ extracted by a neural network. A regressor $f_{\boldsymbol{\theta}}$ parameterized by $\boldsymbol{\theta}$ maps $\mathbf{z}_i$ to a predicted target $\hat{y}_i = f_{\boldsymbol{\theta}}(\mathbf{z}_i)$. Specifically, when $f_{\boldsymbol{\theta}}$ is a linear regressor, which is typically the case in deep neural networks, we have $\hat{y}_i = \boldsymbol{\theta}^\mathsf{T} \mathbf{z}_i$. The encoder and $f_{\boldsymbol{\theta}}$ are trained by minimizing a task-specific regression loss $\mathcal{L}_{reg}$. Typically, the mean-squared error is used, i.e. $\mathcal{L}_{reg} = \frac{1}{N} \sum_{i=1}^N (y_i - \hat{y}_i)^2$.

To formulate regression as a classification problem, the continuous target $y$ is quantized to $K$ classes $y_i^c \in \{1, \cdots, K\}$, and the cross-entropy loss is used to train the encoder and classifiers

$\mathcal{L}_{CE} = -\frac{1}{N} \sum_{i=1}^{N} \log \frac{\exp \boldsymbol{\theta}_{y_i^c}^\mathsf{T} \mathbf{z}_i}{\sum_{j=1}^{K} \exp \boldsymbol{\theta}_j^\mathsf{T} \mathbf{z}_i}$, where $\boldsymbol{\theta}_k$ is the classifier [1] corresponding to the class $k$. The function $d(*, *)$ measures some distance between two points, *e.g.*, Euclidean distance.

## 3.2 ORDINALITY AND TIGHTNESS

This section shows that preserving ordinality tightens the learned representation, and conversely, tightening the representation will help preserve ordinality. A lower $\mathcal{H}(\mathbf{Z}|\mathbf{Y})$ represents a higher compression (Zhang et al., 2024) . The compression is maximized when $\mathcal{H}(\mathbf{Z}|\mathbf{Y})$ is in its minimal ($\mathcal{H}(\mathbf{Z}|\mathbf{Y}) = -\infty$ for differential entropy and $\mathcal{H}(\mathbf{Z}|\mathbf{Y}) = 0$ for discrete entropy).

First, we define the ordinality following (Gong et al., 2022):

**Definition 1** *(Ordinality). The ordinality is perfectly preserved if $\forall i, j, k$, the following holds: $d(y_i, y_j) \leq d(y_i, y_k) \Rightarrow d(\mathbf{z}_i, \mathbf{z}_j) \leq d(\mathbf{z}_i, \mathbf{z}_k)$.*

**Theorem 1** *Let $\mathcal{B}(\mathbf{z}, \epsilon) = \{\mathbf{z}' \in \mathcal{Z} | d(\mathbf{z}, \mathbf{z}') \leq \epsilon\}$ be the closed ball center at $\mathbf{z}$ with radius $\epsilon$. Assume that $\forall(\mathbf{x}, \mathbf{z}, y) \in \mathcal{P}$ and $\forall \epsilon > 0, \exists(\mathbf{x}', \mathbf{z}', y') \in \mathcal{P}$ such that $\mathbf{z}' \in \mathcal{B}(\mathbf{z}, \epsilon)$ and $y' \neq y$. Then if the ordinality is perfectly preserved, $\forall(\mathbf{x}_i, \mathbf{z}_i, y_i), (\mathbf{x}_j, \mathbf{z}_j, y_j) \in \mathcal{P}$, the following hold: $y_i = y_j \Rightarrow d(\mathbf{z}_i, \mathbf{z}_j) = 0$.*

The detailed proof of Theorem 1 is given in Appendix A.1. Theorem 1 states that if the ordinality is perfectly preserved, then the tightness (i.e. $\mathcal{H}(\mathbf{Z}|\mathbf{Y})$) is minimized. This suggests that preserving ordinality will tighten the representations. The assumption in Theorem 1 aligns with the learning target that learning continuously changes representations from continuous targets.

Conversely, if the representations can be correctly mapped to the target and are perfectly tightened, then the representations collapse into a manifold homeomorphic to the target space (e.g., collapse into a single line when the target space is a line) [(Zhang et al., 2024), Proposition 2]. Thus, ordinality will be perfectly preserved locally. Note that reserving ordinality globally constrains the line to be straight, which is not necessary.

## 3.3 REGRESSION TIGHTNESS

Why are additional efforts to emphasize ordinality necessary? In this work, we find that standard deep regressors are weak in their ability to tighten the representations due to the gradient update direction with respect to the representations. Consider a fixed linear regression with a typical regression loss (e.g., MSE, L1), which has the following gradient with respect to $\mathbf{z}_i$:

$$\frac{\partial \mathcal{L}_{\text{reg}}}{\partial \mathbf{z}_i} = \mathcal{L}'_{\text{reg}}(\boldsymbol{\theta}^\mathsf{T}\mathbf{z}_i - y_i)\boldsymbol{\theta}^\mathsf{T}. \tag{1}$$

Here, the direction of $\frac{\partial \mathcal{L}_{\text{reg}}}{\partial \mathbf{z}_i}$ is determined solely by the direction of $\boldsymbol{\theta}$. As such, during learning, all the $\mathbf{z}_i$ are moved in the direction of $\boldsymbol{\theta}$ (or away). This movement can be regarded as a probability density shift (Sonoda & Murata, 2019) , so regression suffers from a weak ability to change probability density in directions perpendicular to $\boldsymbol{\theta}$, which indicates a limited ability to tighten representations in those directions. In other words, regressors can only move $\mathbf{z}_i$ to $S_{y_i}$, but cannot tighten $S_{y_i}$, where $S_{y_i} = \{\mathbf{z} | f_{\boldsymbol{\theta}}(\mathbf{z}) = y_i\}$ is the solution space of $f_{\boldsymbol{\theta}}(\mathbf{z}) = y_i$. More generally, for a differentiable regressor, we have the following:

**Theorem 2** *Assume $f_{\boldsymbol{\theta}}$ is differentiable and $S_{y'_i}$ is a convex set, then $\forall \mathbf{z}'_i, \mathbf{z}'_j \in S_{y'_i}$:*

$$\frac{\partial \mathcal{L}_{reg}}{\partial \mathbf{z}_i}(\mathbf{z}'_i - \mathbf{z}'_j) = 0, \tag{2}$$

*where $y'_i$ is the predicted target of $\mathbf{z}_i$.*

The detailed proof of Theorem 2 is given in Appendix A.2. The regressor $f_{\boldsymbol{\theta}}$ is generally differentiable for gradient backpropagation, and $S_{y_i}$ is commonly a convex set with widely used regressors, such as the linear regressor. Theorem 2 shows that the gradient with respect to the representation will be perpendicular to its solution space and has no effect within the solution space. In other words, with a

---

[1]In this work, a classifier represents a single $\boldsymbol{\theta}_j$ rather than the whole set $\{\boldsymbol{\theta}_j | j \in K\}$.

fixed regressor, the gradient only moves the representations to the corresponding solution space and lags in its ability to tighten the feature space.

In reality, the regressor is not fixed (i.e., updating with training), and the solution space is also changing during training. In the case of a linear regressor, the gradient with respect to $\boldsymbol{\theta}$ over a batch of $b$ samples can be given as:

$$\frac{\partial \mathcal{L}_{\text{reg}}}{\partial \boldsymbol{\theta}} = \frac{1}{b} \sum_{i=1}^{b} \mathcal{L}'_{\text{reg}}(\boldsymbol{\theta}^{\mathsf{T}} \mathbf{z}_i - y_i) \mathbf{z}_i^{\mathsf{T}} = \frac{1}{b} \sum_{i=1}^{b} \mathbf{w}_i \mathbf{z}_i^{\mathsf{T}}. \tag{3}$$

Here, the direction of $\frac{\partial \mathcal{L}_{\text{reg}}}{\partial \boldsymbol{\theta}}$ will tend to be the weighted mean of the direction of $\mathbf{z}_i$. As discussed, the direction of $\mathbf{z}_i$ approaches the direction of $\boldsymbol{\theta}$. Thus, $\mathbf{z}_i$ will distribute around $\boldsymbol{\theta}$ and offset each other, resulting in a limited impact on the direction of $\boldsymbol{\theta}$.

It is worth mentioning that the tightness here is specific to $\mathcal{H}(\mathbf{Z}|\mathbf{Y} = y_i)$ within $S_{y_i}$, which is indirectly related to the predicted results and performance. By contrast, the tightness outside $S_{y_i}$ directly affects the predicted results and potentially plays a more important role.

### 3.4 Comparison in Tightness for Classification

When comparing classification with regression, we find that classification has greater flexibility to tighten representations in diverse directions $\boldsymbol{\theta}_k$, suggesting an ability to better tighten the representation. For the gradient with respect to $\mathbf{z}_i$ over a batch of b samples:

$$\frac{\partial \mathcal{L}_{CE}}{\partial \mathbf{z}_i} = \frac{\partial(-\frac{1}{b}\sum_{i=1}^{b} \boldsymbol{\theta}_{y_i}^{\mathsf{T}} \mathbf{z}_i)}{\partial \mathbf{z}_i} + \frac{\partial(\frac{1}{b}\sum_{i=1}^{b} \log \sum_{j=1}^{K} e^{\boldsymbol{\theta}_j^{\mathsf{T}} \mathbf{z}_i})}{\partial \mathbf{z}_i} = \frac{1}{b} \sum_{i=1}^{b} \left( \sum_{j=1}^{K} p_{ij} \boldsymbol{\theta}_j^{\mathsf{T}} - \boldsymbol{\theta}_{y_i}^{\mathsf{T}} \right) \tag{4}$$

where $p_{ij} = \frac{\exp \boldsymbol{\theta}_j^{\mathsf{T}} \mathbf{z}_i}{\sum_{k=1}^{K} \exp \boldsymbol{\theta}_k^{\mathsf{T}} \mathbf{z}_i}$ is the probability of sample $i$ belonging to class $j$. Here, the direction of $\frac{\partial \mathcal{L}_{CE}}{\partial \mathbf{z}_i}$ is affected by all $\boldsymbol{\theta}_k$, and $\mathbf{z}_i$ will approach $\boldsymbol{\theta}_{y_i}$ with training. In contrast, the direction of $\frac{\partial \mathcal{L}_{Reg}}{\partial \mathbf{z}_i}$ is purely determined by $\boldsymbol{\theta}$. Classification moves $\mathbf{z}_i$ to its corresponding classifier $\boldsymbol{\theta}_{y_i}$ even if the sample is correctly classified. At the same time, regression has no effect on $\mathbf{z}_i$ if it is correctly predicted (i.e., $\frac{\partial \mathcal{L}_{\text{reg}}}{\partial \mathbf{z}_i} = 0$). This suggests that classification has a higher ability to tighten representations in the solution space $S_{y_i}$. Here, $S_{y_i}$ for classification is defined as the set of $\mathbf{z}_i$ that are classified as class $y_i$.

In reality, the classifiers $\boldsymbol{\theta}_k$ are not fixed and are updated with training. The gradient with respect to $k^{\text{th}}$ classifier $\boldsymbol{\theta_k}$ over a batch of b samples is given as:

$$\frac{\partial \mathcal{L}_{CE}}{\partial \boldsymbol{\theta}_k} = -\frac{1}{b} \sum_{i: y_i = k} \mathbf{z}_i^{\mathsf{T}} + \frac{1}{b} \sum_{i=1}^{b} \frac{e^{\boldsymbol{\theta}_k^{\mathsf{T}} \mathbf{z}_i^{\mathsf{T}}}}{\sum_{j=1}^{K} e^{\boldsymbol{\theta}_j^{\mathsf{T}} \mathbf{z}_i}} \mathbf{z}_i^{\mathsf{T}} = \frac{1}{b} \sum_{i=1}^{b} (p_{ik} - \delta_{y_i,k}) \mathbf{z}_i^{\mathsf{T}} \tag{5}$$

where $p_{ik} = \frac{e^{\boldsymbol{\theta}_k^{\mathsf{T}} \mathbf{z}_i}}{\sum_{j=1}^{K} e^{\boldsymbol{\theta}_j^{\mathsf{T}} \mathbf{z}_i}}$ is the probability of sample $i$ belongs to the class $k$, and $\delta_{y_i,k}$ is the Kronecker delta function. For classification, the direction of $\boldsymbol{\theta}_k$ will biased toward $\mathbf{z}_i$ with respect to the class $k$ , while $\mathbf{z}_i$ will also bias towards its corresponding classifier. In contrast, for regression, the direction of $\boldsymbol{\theta}$ will tend to be the weighted mean of the direction of $\mathbf{z}_i$. Thus, the effect of many $\mathbf{z}_i$ on the direction of $\boldsymbol{\theta}$ will offset each other and have a limited impact. As a result, changes in the directions of $\boldsymbol{\theta}_k$ are generally greater than the change of the $\boldsymbol{\theta}$ direction in regression, and therefore classification can move $\mathbf{z}$ more flexible and thus potentially better tighten the representation.

## 4 Method

Our analysis in Sec. 3 inspires us to tighten the regression representations. To this aim, we introduce the Multiple Target (**MT**) strategy and the Regression Optimal Transport Regularizer (**ROT-Reg**) to tighten the representations globally (i.e., $\min_{\mathbf{Z}} \mathcal{H}(\mathbf{Z}|\mathbf{Y})$) and locally (i.e., $\min_{\mathbf{Z}} \mathcal{H}(\mathbf{Z}|\mathbf{Y} = y_i), \mathbf{Z} \in \mathcal{B}(\mathbf{z}_i, \epsilon)$, $\epsilon$ control the degree of locality). Inspired by the effect of multiple classifiers in classification, the MT strategy introduces additional targets as constraints to compress the representations. For ROT-Reg, we exploit it to encourage representations to have local structures similar to the targets, which implicitly tightens the representations.

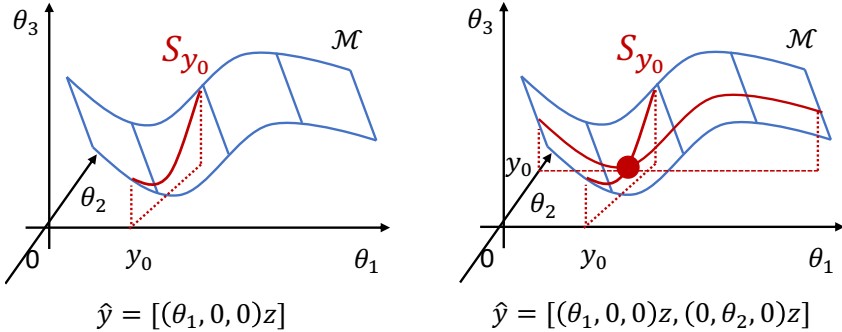

$$\hat{y} = [(\theta_1, 0, 0)z] \qquad\qquad \hat{y} = [(\theta_1, 0, 0)z, (0, \theta_2, 0)z]$$

Figure 1: Illustration of the MT strategy. Changing the target from $y$ to $[y, y]$ will introduce an additional regressor to predict the additional $y$. The original solution space $S_{y_0}$ is a line in the feature manifold. The additional $y$ introduces a new constraint, tightening $S_{y_0}$ from a line to a point.

### 4.1 Target Space with Extra Dimensions Better Tighten the Feature Space

Our analysis in Sec. 3.4 suggests that classification outperforms regression in its ability to compress representations in multiple directions, resulting from multiple classifiers. Inspired by this, we introduce a simple yet efficient strategy, which adds extra dimensions for the target space to bring in extra regressors as constraints. Here, the additional regressors have a similar effect as individual classifiers. As shown in Figure 1, the additional constraints will result in a lower-dimensional $S_{y_i}$, which indicates higher compression. The number of additional targets depends on the intrinsic dimension of the feature manifold. In our Multiple Targets strategy, the final predicted target is the average over the multiple predicted targets:

$$\hat{y}_i = \frac{1}{M} \sum_{t=1}^{M} \hat{y}_i^t, \tag{6}$$

where $M$ is the number of the total target dimension and $\hat{y}_i^t$ is the $t^{th}$ predicted target.

### 4.2 Regression Optimal Transport Regularizer (ROT-Reg)

The MT strategy tightens the representations globally through additional regressors. We propose to further tighten the representations locally. Specifically, we preserve the local similarity relations between the target and representation space. The local similarities are characterized by a self entropic optimal transport model (Yan et al., 2024; Landa et al., 2021). The model determines the optimal plan is to move a set of samples to the set itself with minimal transport costs, while each sample cannot be moved to itself.

Formally, Given a set $S = \{\mathbf{s}_1, \ldots, \mathbf{s}_n\}$, the corresponding weight vector $\mathbf{p} = \mathbb{R}^n$ reflects how many masses the samples have, where the weights simplify the simplex constraint $\sum_{i=1}^{n} p_i = 1$. Usually, one can easily implement $\mathbf{p}$ as a uniform distribution, i.e., $p_i = \frac{1}{n}, \forall i \in [n]$. $C_{ij}^{\mathbf{S}}$ is the transport cost between $\mathbf{s}_i$ and $\mathbf{s}_j$, which is generally adopted as the Euclidean distance between the samples, and $T_{ij}^{\mathbf{S}}$ indicates how many masses are transported from the locations of $\mathbf{s}_i$ to $\mathbf{s}_j$. The self entropic optimal transport is defined as follows:

$$\mathcal{T}(\mathbf{S}) = \arg\min_{\mathbf{T}} \langle \mathbf{C^S}, \mathbf{T} \rangle + \gamma \Omega(\mathbf{T}) \tag{7}$$

$$\text{s.t. } \mathbf{T}\mathbf{1}_n = \mathbf{p}, \mathbf{T}^\top \mathbf{1}_n = \mathbf{p}, T_{ii} = 0 \ \forall i \in [n], \tag{8}$$

where $\gamma$ is a trade-off parameter, and $\Omega(\mathbf{T}) = \sum_{i=1}^{n} \sum_{j=1}^{n} T_{ij}^{\mathbf{S}} \log T_{ij}^{\mathbf{S}}$ is the negative entropic regularization, which is used to smoothen the solution and speed up the optimization (Cuturi, 2013).

Given the solution $\widetilde{\mathbf{T}}^{\mathbf{S}}$ minimizing the above objective, the element $T_{ij}^{\mathbf{S}}$ measures the similarity relation between samples $\mathbf{s}_i$ and $\mathbf{s}_j$, since two samples with a large distance $C_{ij}^{\mathbf{S}}$ will induce a small transport mass $T_{ij}^{\mathbf{S}}$ between them. As a result, the optimal total transport cost $\langle \mathbf{C}^s, \widetilde{\mathbf{T}}^{\mathbf{S}} \rangle$ reflects the tightness of the samples.

Motivated by this, we employ the self optimal transport model to capture local similarity relations of target and representation spaces, respectively, and encourage a relation consistency between two spaces. In specific, we first construct a self optimal transport model on the target space to obtain $\widetilde{\mathbf{T}}^{\mathbf{Y}} = \mathcal{T}(\mathbf{Y})$, which describes the local similarity relations between the regression targets. After that, we learn regression representations $\mathbf{Z}$ such that the corresponding optimal transport matrix $\widetilde{\mathbf{T}}^{\mathbf{Z}} = \mathcal{T}(\mathbf{Z})$ is consistent with $\widetilde{\mathbf{T}}^{\mathbf{Y}}$, which is achieved by the following loss function

$$\mathcal{L}_{ot} = \left| \langle \mathbf{C}^{\mathbf{Z}}, \widetilde{\mathbf{T}}^{\mathbf{Y}} \rangle - \langle \mathbf{C}^{\mathbf{Z}}, \widetilde{\mathbf{T}}^{\mathbf{Z}} \rangle \right|. \tag{9}$$

ROT-Reg is updating $\mathbf{C}^{\mathbf{Z}}$ through gradient backpropagation to minimize $\mathcal{L}_{ot}$. In contrast, simply minimizing the gap of $\widetilde{\mathbf{T}}^{\mathbf{Y}}$ and $\widetilde{\mathbf{T}}^{\mathbf{Z}}$ can introduce optimization challenges, as the two matrices are obtained iteratively through the Sinkhorn algorithm rather than simply through gradient backpropagation. In addition, directly minimizing $||\mathbf{C}^{\mathbf{Z}} - \mathbf{C}^{\mathbf{Y}}||_F$ imposes an overly strict constraint on the feature manifold, forcing it to become identical to the target space, which is unnecessary.

It is worth mentioning that $\gamma$ controls the 'smooth' of the transport plan $\mathbf{T}$, and determines the degree of locality. When $\gamma = 0$, $\mathbf{T}$ will approach the minimal spanning tree (i.e., only transports mass to its nearest neighbor), and $\mathcal{L}_{ot}$ will encourage the representations to have the same minimal spanning tree to the targets, which is shown to be a strategy to preserve the topology of the target space (Moor et al., 2020). In fact, the topological auto-encoder (Moor et al., 2020) preserves topological information in this way. Compared to topology autoencoder, $\mathcal{L}_{ot}$ captures more local structures of targets when $\gamma > 0$. The final loss $\mathcal{L}_f$ sums the task-specific loss $\mathcal{L}_t$ and the regularizer with a trade-off hyper-parameter $\lambda$ :

$$\mathcal{L}_f = \mathcal{L}_t + \lambda \mathcal{L}_{ot}, \tag{10}$$

## 5 EXPERIMENTS

We experiment on three deep regression tasks: age estimation, depth estimation, and coordinate prediction and compare with RankSim (Gong et al., 2022), Ordinal Entropy (OE) (Zhang et al., 2023), and PH-Reg (Zhang et al., 2024). RankSim explicitly preserves ordinality to serve as the ordinality baseline. OE leverages classification for better regression representations and serves as a regression baseline. PH-Reg preserves the topological structure of the target space by the Topological autoencoder (Moor et al., 2020) and tightens the representation by Birdal's regularizer (Birdal et al., 2021), serving as a topology baseline. More details are given in Appendix B.1.

### 5.1 REAL-WORLD DATASETS: AGE ESTIMATION AND DEPTH ESTIMATION

For age estimation, we use AgeDB-DIR (Yang et al., 2021) and evaluate using Mean Absolute Error (MAE) as the evaluation metric. $\gamma$ and $\lambda$ are set to $0.1$ and $100$, respectively. For depth estimation, we use NYUD2-DIR (Yang et al., 2021) and evaluate using the root mean squared error (RMSE) and the threshold accuracy $\delta_1$ as the evaluation metrics. $\gamma$ and $\lambda$ are set to $0.05$ and $10$, respectively. We set the total target dimension $M$ to be $8$ for both tasks. Both AgeDB-DIR and NYUD2-DIR contain three disjoint subsets (i.e., Many, Med, and Few) divided from the whole set. We exploit the regression baseline models of (Yang et al., 2021), which use ResNet-50 (He et al., 2016) as the backbone, and follow their setting for both tasks.

Tables 1 and 2 show results on age estimation and depth estimation respectively. Both the Multiple Targets strategy (**MT**) and $\mathcal{L}_{ot}$ improve regression performance, and combining both further boosts performance. Specifically, combining both achieves $0.52$ overall improvements (i.e. ALL) on age estimation, and a $0.156$ reduction of RMSE on depth estimation.

### 5.2 $\mathcal{L}_{ot}$ PRESERVES THE LOCAL SIMILARITY RELATIONSHIPS

The effectiveness of $\mathcal{L}_{ot}$ is verified with the coordinate prediction task from Zhang et al. (2024). This task predicts data coordinates sampled from manifolds such as Mammoth, Torus, and Circle, which have different topologies. The inputs are noisy data samples and the goal is to recover the true data coordinates. Figure 2 shows that $\mathcal{L}_{ot}$ successfully preserves the similarity relationships of the targets, resulting in a feature manifold similar to the targets. Quantitative comparisons in Table 3 indicate

Table 1: Quantitative comparison (MAE) on AgeDB-DIR. We report results as mean ± standard deviation over 10 runs. **Bold** numbers indicate the best performance.

| Method | ALL | Many | Med. | Few |
|---|---|---|---|---|
| Baseline | $7.80 \pm 0.12$ | $6.80 \pm 0.06$ | $9.11 \pm 0.31$ | $13.63 \pm 0.43$ |
| + RankSim | $7.62 \pm 0.13$ | $6.70 \pm 0.10$ | $8.90 \pm 0.33$ | $12.74 \pm 0.48$ |
| + OE | $7.65 \pm 0.13$ | $6.72 \pm 0.09$ | $8.77 \pm 0.49$ | $13.28 \pm 0.73$ |
| +PH-Reg | $7.32 \pm 0.09$ | $6.50 \pm 0.15$ | $8.38 \pm 0.11$ | $12.18 \pm 0.38$ |
| + MT | $7.67 \pm 0.06$ | $6.72 \pm 0.08$ | $8.87 \pm 0.13$ | $13.36 \pm 0.16$ |
| +$\mathcal{L}_{oe}$ | $7.36 \pm 0.08$ | $6.55 \pm 0.07$ | $8.40 \pm 0.14$ | $12.14 \pm 0.33$ |
| + MT + $\mathcal{L}_{oe}$ | $\mathbf{7.28 \pm 0.05}$ | $\mathbf{6.52 \pm 0.10}$ | $\mathbf{8.26 \pm 0.19}$ | $\mathbf{11.86 \pm 0.24}$ |

Table 2: Quantitative comparison on NYUD2-DIR.

| Method | RMSE ↓ | | | | $\delta_1$ ↑ | | | |
|---|---|---|---|---|---|---|---|---|
| | ALL | Many | Med. | Few | ALL | Many | Med. | Few |
| Baseline | 1.477 | 0.591 | 0.952 | 2.123 | 0.677 | 0.777 | 0.693 | 0.570 |
| +RankSim | 1.522 | **0.565** | 0.889 | 2.213 | 0.666 | 0.791 | 0.735 | 0.513 |
| +OE | 1.419 | 0.671 | 0.925 | 2.005 | 0.668 | 0.727 | 0.702 | 0.596 |
| +PH-Reg | 1.450 | 0.789 | 0.911 | 2.002 | 0.620 | 0.621 | 0.680 | 0.596 |
| + MT | 1.367 | 0.605 | **0.854** | 1.952 | **0.715** | **0.776** | **0.759** | 0.636 |
| +$\mathcal{L}_{ot}$ | 1.353 | 0.654 | 0.934 | 1.899 | 0.689 | 0.736 | 0.697 | 0.638 |
| + MT + $\mathcal{L}_{ot}$ | **1.321** | 0.685 | 0.951 | **1.829** | 0.701 | 0.731 | 0.689 | **0.675** |

Table 3: Results ($\mathcal{L}_{\mathrm{mse}}$) on the coordinate prediction dataset. We report results as mean ± standard deviation over 10 runs. **Bold** numbers indicate the best performance.

| Method | Mammoth | Torus | Circle |
|---|---|---|---|
| Baseline | $211 \pm 55$ | $3.01 \pm 0.11$ | $0.154 \pm 0.006$ |
| + InfDrop | $367 \pm 50$ | $2.05 \pm 0.04$ | $0.093 \pm 0.003$ |
| + OE | $187 \pm 88$ | $2.83 \pm 0.07$ | $0.114 \pm 0.007$ |
| +Topological Autoencoder | $80 \pm 61$ | $0.95 \pm 0.05$ | $0.036 \pm 0.004$ |
| + PH-Reg | $\mathbf{49 \pm 27}$ | $\mathbf{0.61 \pm 0.05}$ | $0.013 \pm 0.008$ |
| + MT | $174 \pm 76$ | $2.99 \pm 0.11$ | $0.152 \pm 0.005$ |
| + $\mathcal{L}_{ot}$ | $87 \pm 26$ | $0.77 \pm 0.02$ | $0.010 \pm 0.001$ |
| + MT+ $\mathcal{L}_{ot}$ | $76 \pm 53$ | $0.75 \pm 0.03$ | $\mathbf{0.010 \pm 0.001}$ |

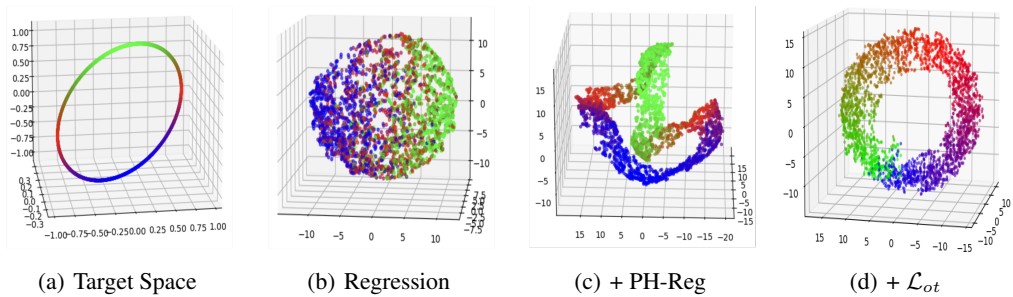

    (a) Target Space      (b) Regression      (c) + PH-Reg      (d) + $\mathcal{L}_{ot}$

Figure 2: Visualization of the feature manifolds, which shows that $\mathcal{L}_{ot}$ preserves the local similarity relationships of the target space.

that $\mathcal{L}_{oe}$ performs similarly to PH-Reg, specifically designed to preserve similarity relationships. However, the Multiple Targets (MT) strategy has a limited impact in this context, likely because the target space is three-dimensional, providing sufficient constraints for the feature manifold.

Table 4: Correlation between feature and label similarities. Results are mean $\pm$ std dev over 10 runs.

| Method | RMSE (ALL) | Cosine Distance | | | Euclidean Distance | | |
|---|---|---|---|---|---|---|---|
| | | Spearman's ↑ | Kendall's ↑ | volume | Spearman's ↑ | Kendall's ↑ | volume |
| Baseline | 1.477 | $0.39 \pm 0.15$ | $0.27 \pm 0.11$ | $0.573 \pm 0.071$ | $0.35 \pm 0.14$ | $0.24 \pm 0.10$ | $7.72 \pm 0.92$ |
| + RankSim | 1.522 | $0.09 \pm 0.04$ | $0.06 \pm 0.03$ | $0.000 \pm 0.000$ | $0.60 \pm 0.16$ | $0.44 \pm 0.13$ | $4.26 \pm 1.29$ |
| + MT | 1.367 | $0.49 \pm 0.14$ | $0.34 \pm 0.10$ | $0.492 \pm 0.047$ | $0.47 \pm 0.11$ | $0.33 \pm 0.08$ | $6.56 \pm 1.18$ |
| + $\mathcal{L}_{ot}$ | 1.353 | $0.48 \pm 0.16$ | $0.34 \pm 0.12$ | $0.010 \pm 0.003$ | $0.42 \pm 0.15$ | $0.29 \pm 0.11$ | $5.57 \pm 0.77$ |
| + MT + $\mathcal{L}_{ot}$ | 1.321 | $0.64 \pm 0.09$ | $0.46 \pm 0.08$ | $0.006 \pm 0.002$ | $0.61 \pm 0.11$ | $0.44 \pm 0.09$ | $4.16 \pm 0.51$ |

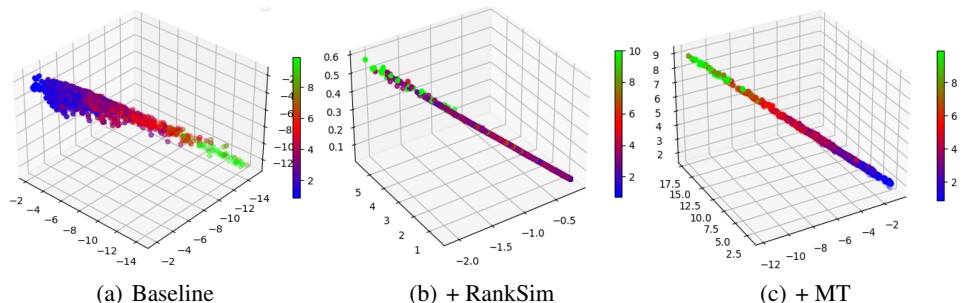

| (a) Baseline | (b) + RankSim | (c) + MT |
|---|---|---|

Figure 3: Visualizations of the feature manifold on NYUD2-DIR for depth estimation. Preserving the ordinality (+ RankSim) has an effect similar to MT, which explicitly tightens the representations.

## 5.3 TIGHTNESS AND ORDINALITY AFFECT EACH OTHER

**Compression for a better ordinality**. We examine the impact of tightness on the ordinality. Table 4 presents the Spearman's rank correlation coefficient (Spearman, 1961) and Kendall rank correlation coefficient (Kendall, 1938) between the feature similarities (based on Cosine distance and Euclidean distance) and the label similarities. The two correlation coefficients measure how well ordinality is preserved. Since tightness compresses the feature manifold, reducing its volume, we use volume as a proxy for tightness, and the volume is approximated by the mean of the similarities between samples. The experiments are conducted on NYUD2-DIR, we randomly sample 1000 pixels from a batch of 8 test images. The label similarities are calculated as the Euclidean distances between the 1000 pixels, while the corresponding feature similarities are the distances between their corresponding representations. The results in Table 4 show standard regression fails to preserve the ordinality, while MT and $\mathcal{L}_{ot}$ both improve the ordinality, although they are designed to tighten the representations. Combining both has a similar effect on preserving ordinality as RankSim, which is specifically designed for this purpose. The lower volumes of our method compared to the baseline indicate that the feature manifold is more compressed. We provide the visualization of the feature similarities in Appendix C.1.

**Ordinality for a better compression**. To further verify that preserving ordinality leads to better compression, we visualize the feature manifold of the depth estimation task in 3D space. This is done by changing the last hidden layer's feature space to three dimensions. As shown in Figure 3, explicitly preserving the ordinality (i.e., +RankSim) compresses the feature manifold into a thin line, which shows a similar effect to explicitly tightening the representations (i.e., +MT).

## 5.4 TIGHTNESS OF REGRESSION

Our theoretical analysis in Sec. 3 focuses on the gradient direction of representations. However, in reality, the neural network updates its parameters to update the representations indirectly. Here, we verify our analysis by visualizing the update of $\mathbf{z}$ and $\boldsymbol{\theta}$ in the depth estimation task.

**The update of $\mathbf{z}$**. We change the last hidden layer's feature space to 2 dimensional for visualization. We randomly sample 1000 pixels from a batch of 8 images in the NYUD2-DIR test set to visualize the feature manifold. Figure 4(a) displays the feature manifolds at epoch 1 (blue dots) and epoch 10 (the final epoch, red dots), the corresponding pixel representations are connected by black arrows. Aligned with our theoretical analysis, the directions of the representation updates follow the direction of $\boldsymbol{\theta}$. To verify this quantitatively, we calculate the principal component of the update directions using

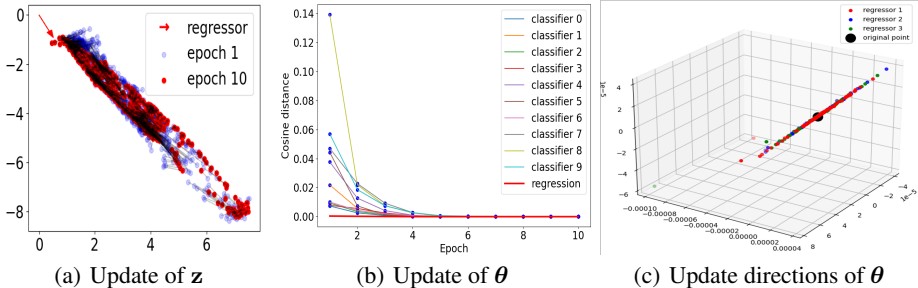

|(a) Update of **z** | (b) Update of $\boldsymbol{\theta}$ | (c) Update directions of $\boldsymbol{\theta}$ |

Figure 4: (a) Visualization of the **z** update, which aligns with $\theta$, (b) $\boldsymbol{\theta}$ update, which is steady through the training process, (c) the updating directions of $\boldsymbol{\theta}$s, which distributed along a line, with the original as the center.

PCA. We find that the cosine distance between this principal component and $\boldsymbol{\theta}$ is very small (0.03), indicating that the updating directions of representations from the beginning to the end of training follow the direction of $\boldsymbol{\theta}$. The visualization also shows that the feature manifold tightened limited in the direction perpendicular to $\boldsymbol{\theta}$ throughout the training. The visualizations of feature manifolds at each epoch are provided in Appendix C.2, which reveals that the tightening effect in the direction perpendicular to $\boldsymbol{\theta}$ between adjacent epochs is even smaller.

**The update of $\boldsymbol{\theta}$.** As discussed in Sec. 3, the effect of $\mathbf{z}_i$ on the direction of $\boldsymbol{\theta}$ tends to offset each other and results in a limited impact, while changes in the directions of $\boldsymbol{\theta}_k$ in classification are generally greater. Here we quantitatively verify this by calculating the cosine distances between $\tilde{\theta}$ and $\boldsymbol{\theta}$ at each epoch from 1 to 10 (final epoch), where $\tilde{\theta}$ represents $\boldsymbol{\theta}$ at epoch 10. We also convert this regression task into a classification task by uniformly discretizing the target range into 10 classes, and monitoring the change of $\boldsymbol{\theta}_k$ in the same way. As shown in Figure 4(b), the changes in $\boldsymbol{\theta}_k$ are all larger than the changes in $\boldsymbol{\theta}$. The maximum cosine distance between $\boldsymbol{\theta}$ at different epochs is very small (i.e., 0.0004), which also verified the limited change of $\boldsymbol{\theta}$.

**Multiple $\boldsymbol{\theta}$s.** Adding additional $\boldsymbol{\theta}$s (our MT strategy) , with random initialization, does not change the update speed of $\boldsymbol{\theta}$ (see Figure 7 in the Appendix C.3). The update directions of all $\boldsymbol{\theta}$s are even aligned. Let $v_{\boldsymbol{\theta}}^i = \boldsymbol{\theta}^{i+1} - \boldsymbol{\theta}^i$ be the updating vector of $\boldsymbol{\theta}$ in iteration $i$. Figure 4(c) plots the set of points $\{v_{\boldsymbol{\theta}}^i | i = 500k, k \in \mathcal{Z}, 0 \le k \le 100\}$ for three $\boldsymbol{\theta}$s. This visualizes the change of $\boldsymbol{\theta}$s throughout the training process. The three $\boldsymbol{\theta}$s are distributed along a line, with the original as the center. When we calculate the principle components of $\{v_{\boldsymbol{\theta}}\}$ for the three $\boldsymbol{\theta}$s using PCA, the maximum cosine distances between the three principle components are less than $1e-4$, which quantitatively shows the updating directions of all $\boldsymbol{\theta}$s are in the same direction. As shown in Eq. 3, $v_{\boldsymbol{\theta}}$ is the weighted mean of a batch of **z**. Different $\boldsymbol{\theta}$ leads to different magnitudes of the weight mean while the directions remain steady. It is worth mentioning that the multiple $\boldsymbol{\theta}$s do not collapse to a single $\boldsymbol{\theta}$ in reality, although their updating directions are the same. This is because $\boldsymbol{\theta}$s are randomly initialized, and their directions remain nearly identical during training (see Figure 4(b)), due to the three reasons: 1) The magnitude of $\frac{\partial \mathcal{L}_{\text{reg}}}{\partial \boldsymbol{\theta}}$ is 'scaled' by $\mathbf{w}_i$, since $\mathbf{w}_i$ often follows a Gaussian distribution centered at the origin, as assumed in models like Bayesian linear regression. When **w** and **z** are independent or weak dependent, $\mathbb{E}[\mathbf{w}_i \mathbf{z}_i]$ will approach 0 and causing $\frac{\partial \mathcal{L}_{\text{reg}}}{\partial \boldsymbol{\theta}}$ be 'scaled' to 0. 2) According to the central limit theorem, the updates of $\boldsymbol{\theta}$ follow a Gaussian distribution. This causes partial offsets between updates and results in a reduced accumulated effect. In addition, we empirically observe that the mean of this Gaussian distribution approaches 0 (see Figure 4(c)), indicating that **w** and **z** are independent or weakly dependent. 3) The effect of $\mathbf{z}_i$ on the direction of $\frac{\partial \mathcal{L}_{\text{reg}}}{\partial \boldsymbol{\theta}}$ over a batch of samples offsets each other, resulting in the stability of the direction of $\boldsymbol{\theta}$ throughout training. This occurs because $\mathbf{z}_i$ tends to be distributed around $\boldsymbol{\theta}$. More details can be found in Appendix C.3.

## 5.5 ABLATION STUDY

We conduct the ablation study on AgeDB-DIR for age estimations. The results are given in Table 5.

Table 5: Ablation study on AgeDB-DIR for age estimation. We report MAE mean ± standard deviation **over 10 runs**, and the default $\lambda$, $\gamma$ and $M$ are set to 100, 0.1 and 8, respectively.

| Method | ALL | Many | Med. | Few |
|---|---|---|---|---|
| Baseline | $7.80 \pm 0.12$ | $6.80 \pm 0.06$ | $9.11 \pm 0.31$ | $13.63 \pm 0.43$ |
| + $\mathcal{L}_{oe}$ | | | | |
| $\lambda = 1$ | $7.68 \pm 0.08$ | $6.75 \pm 0.11$ | $8.81 \pm 0.19$ | $13.38 \pm 0.37$ |
| $\lambda = 10$ | $7.55 \pm 0.05$ | $6.64 \pm 0.07$ | $8.71 \pm 0.12$ | $12.88 \pm 0.35$ |
| $\lambda = 100$ | $7.36 \pm 0.08$ | $6.55 \pm 0.07$ | $8.40 \pm 0.14$ | $12.14 \pm 0.33$ |
| $\lambda = 1000$ | $8.80 \pm 0.19$ | $7.17 \pm 0.10$ | $11.32 \pm 0.48$ | $17.26 \pm 0.53$ |
| $\gamma = 0.1$ | $7.36 \pm 0.08$ | $6.55 \pm 0.07$ | $8.40 \pm 0.14$ | $12.14 \pm 0.33$ |
| $\gamma = 1$ | $7.47 \pm 0.12$ | $6.61 \pm 0.08$ | $8.57 \pm 0.31$ | $12.55 \pm 0.49$ |
| $\gamma = 10$ | $7.51 \pm 0.07$ | $6.63 \pm 0.08$ | $8.60 \pm 0.25$ | $12.75 \pm 0.31$ |
| + MT | | | | |
| M=2 | $7.72 \pm 0.11$ | $6.77 \pm 0.07$ | $8.92 \pm 0.20$ | $13.37 \pm 0.50$ |
| M=4 | $7.74 \pm 0.06$ | $6.77 \pm 0.10$ | $8.96 \pm 0.13$ | $13.52 \pm 0.41$ |
| M=8 | $7.67 \pm 0.06$ | $6.72 \pm 0.08$ | $8.87 \pm 0.13$ | $13.36 \pm 0.16$ |
| M=16 | $7.70 \pm 0.11$ | $6.73 \pm 0.13$ | $8.93 \pm 0.25$ | $13.44 \pm 0.36$ |
| M=32 | $7.71 \pm 0.08$ | $6.74 \pm 0.10$ | $8.96 \pm 0.32$ | $13.40 \pm 0.32$ |
| noise | $8.00 \pm 0.23$ | $6.91 \pm 0.20$ | $9.43 \pm 0.33$ | $14.36 \pm 0.67$ |

Table 6: Time and memory consumption.

| Method | Time (mins) | Memory(MB) |
|---|---|---|
| Baseline | 65 | 14433 |
| + MT | 70 | 14457 |
| + $\mathcal{L}_{ot}$ | 74 | 14587 |
| + MT + $\mathcal{L}_{ot}$ | 82 | 14689 |

**Hyperparameter $\lambda$, $\gamma$.** We keep $\lambda$, $\gamma$ at their default value 100, 0.1, and vary each individually to examine their impact. As shown in Table 5, The MAE (ALL) decreases consistently as $\lambda$ increases. However, it overtakes the task-specific learning target when set too high (e.g., 1000) and decreases the performance. For the $\gamma$, the MAE (ALL) decreases consistently as $\gamma$ decreases. However, we empirically find that setting $\gamma$ (e.g., 0.01) too low will easily result in NaN values when calculating the transport matrixes using the Sinkhorn algorithm (Cuturi, 2013). We thus set $\gamma = 0.1$.

**Number of the total targets $M$.** As shown in Table 5, the performance generally improves with the increase of $M$, when $M \leq 8$, and stays steady when $M$ increases further. The primary factor affecting the selection of $M$ is the intrinsic dimension of the feature manifold, which determines how many additional constraints (i.e., $M - 1$) are required to compress the manifold. The range of $\mathbf{Y}$ has a limited impact on the selection of $M$, since $M$ equals 8 works well in NYUD2-DIR ($y \in [0.7, 10]$) and AgeDB-DIR ( $y \in [0, 101]$).

**Mean $\hat{y}$ vs. a single $\hat{y}$.** We verify the strategy of the mean operation in MT (see Eq. 6), which potentially brings in an ensemble effect. We find that $\hat{y}_i^t$ are very similar for all $t$. For a model with $M$, the MAE(ALL) results calculated by $\hat{y}_i^t$ for $t \in T$ are with mean equal to 7.579 and standard deviation 0.0003. Thus, the improvement of MT is not due to the ensemble effect, and the mean operation is optional.

**Additional $y$ vs. noise.** Adding additional targets as noise, as shown in Table 5, does not work.

**Time and memory consumption.** We monitor the time and memory consumption for training a model from the beginning to the end with a batch size equal to 128. Table 6 shows the added memory is negligible(1.7%), and the added time is limited (17 min).

## 6 CONCLUSION

In this paper, for the regression task, we provide a theoretical analysis that suggests preserving ordinality enhances the representation tightness, and regression suffers from a weak ability to tighten the representations. Motivated by classification and the self entropic optimal transport, we introduce a simple yet effective method to tighten regression representations.

**Acknowledgement**. This research / project is supported by the Ministry of Education, Singapore, under the Academic Research Fund Tier 1 (FY2022), National Natural Science Foundation of China (62206061, U24A20233), Guangdong Basic and Applied Basic Research Foundation (2024A1515011901), Guangzhou Basic and Applied Basic Research Foundation (2023A04J1700).

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

## A APPENDIX

### A.1 PROOF OF THEOREM 1

**Theorem 1** *Let $\mathcal{B}(\mathbf{z}, \epsilon) = \{\mathbf{z}' \in \mathcal{Z} | d(\mathbf{z}, \mathbf{z}') \leq \epsilon\}$ be the closed ball center at $\mathbf{z}$ with radius $\epsilon$. Assume that $\forall(\mathbf{x}, \mathbf{z}, y) \in \mathcal{P}$ and $\forall \epsilon > 0, \exists(\mathbf{x}', \mathbf{z}', y') \in \mathcal{P}$ such that $\mathbf{z}' \in \mathcal{B}(\mathbf{z}, \epsilon)$ and $y' \neq y$. Then if the ordinality is perfectly preserved, $\forall(\mathbf{x}_i, \mathbf{z}_i, y_i), (\mathbf{x}_j, \mathbf{z}_j, y_j) \in \mathcal{P}$, the following hold: $y_i = y_j \Rightarrow d(\mathbf{z}_i, \mathbf{z}_j) = 0$.*

**Proof**

$$d(\mathbf{z}_i, \mathbf{z}_j) = d(\mathbf{z}_i - \mathbf{z}_k + \mathbf{z}_k - \mathbf{z}_j) \tag{11}$$
$$\leq d(\mathbf{z}_i - \mathbf{z}_k) + d(\mathbf{z}_k - \mathbf{z}_j), \tag{12}$$

where $\mathbf{z}_k \in \mathcal{B}(\mathbf{z}, \epsilon)$. Since $d(y_k, y_j) \leq d(y_k, y_i)$, and the ordinality is perfectly preserved, we have:

$$d(\mathbf{z}_k - \mathbf{z}_j) \leq d(\mathbf{z}_i - \mathbf{z}_k). \tag{13}$$

Thus:

$$0 \leq d(\mathbf{z}_i, \mathbf{z}_j) \leq 2d(\mathbf{z}_i - \mathbf{z}_k) \leq 2\epsilon. \tag{14}$$

Let $\epsilon \to 0$, the result follows. $\qquad \square$

### A.2 PROOF OF THEOREM 2

We first give a lemma:

**Lemma 1** *Let $S_y = \{\mathbf{z} | g(\mathbf{z}) = y\}$ be a convex set, where $\mathbf{z} \in \mathbb{R}^n$ is the representation, $y$ is the target and $g$ is the regressor. Assume $g$ is differentiable, then $\forall \mathbf{z}_k, \mathbf{z}_i, \mathbf{z}_j \in S_y$, we have:*

$$\nabla g(\mathbf{z}_k)(\mathbf{z}_i - \mathbf{z}_j) = 0. \tag{15}$$

**Proof** Let $\mathbf{z}_k^\epsilon = (1 - \epsilon)\mathbf{z}_k + \epsilon \mathbf{z}_i$, where $\epsilon \in [0, 1]$. Since $g$ is differentiable, using Taylor expansion, we have:

$$g(\mathbf{z}_k^\epsilon) = g((1 - \epsilon)\mathbf{z}_k + \epsilon \mathbf{z}_i) \tag{16}$$
$$= g(\mathbf{z}_k + \epsilon(\mathbf{z}_i - \mathbf{z}_k)) \tag{17}$$
$$= g(\mathbf{z}_k) + \epsilon \nabla g(\mathbf{z}_k)(\mathbf{z}_i - \mathbf{z}_k) + o(\epsilon). \tag{18}$$

Since $S_y$ is a convex set, we have $\mathbf{z}_k^\epsilon \in S_y$. Thus:

$$g(\mathbf{z}_k^\epsilon) = g(\mathbf{z}_k) + \epsilon \nabla g(\mathbf{z}_k)(\mathbf{z}_i - \mathbf{z}_k) + o(\epsilon) \tag{19}$$
$$y = y + \epsilon \nabla g(\mathbf{z}_k)(\mathbf{z}_i - \mathbf{z}_k) + o(\epsilon) \tag{20}$$
$$\frac{o(\epsilon)}{\epsilon} = \nabla g(\mathbf{z}_k)(\mathbf{z}_k - \mathbf{z}_i). \tag{21}$$

Let $\epsilon \to 0$:

$$\nabla g(\mathbf{z}_k)(\mathbf{z}_k - \mathbf{z}_i) = \lim_{\epsilon \to 0} \frac{o(\epsilon)}{\epsilon} = 0. \tag{22}$$

Similarly, we have:

$$\nabla g(\mathbf{z}_k)(\mathbf{z}_k - \mathbf{z}_j) = 0. \tag{23}$$

Combining the two equations above, we have:

$$\nabla g(\mathbf{z}_k)(\mathbf{z}_i - \mathbf{z}_j) = \nabla g(\mathbf{z}_k)(\mathbf{z}_i - \mathbf{z}_k + \mathbf{z}_k - \mathbf{z}_j) \tag{24}$$
$$= \nabla g(\mathbf{z}_k)(\mathbf{z}_i - \mathbf{z}_k) + \nabla g(\mathbf{z}_k)(\mathbf{z}_k - \mathbf{z}_j) \tag{25}$$
$$= 0. \tag{26}$$

$$\square$$

**Theorem 2** *Assume $f_{\boldsymbol{\theta}}$ is differentiable and $S_{y_i'}$ is a convex set, then $\forall \mathbf{z}_i', \mathbf{z}_j' \in S_{y_i'}$:*

$$\frac{\partial \mathcal{L}_{reg}}{\partial \mathbf{z}_i}(\mathbf{z}_i' - \mathbf{z}_j') = 0, \tag{27}$$

*where $y_i'$ is the predicted target of $\mathbf{z}_i$.*

**Proof**

$$\frac{\partial \mathcal{L}_{\text{reg}}}{\partial \mathbf{z}_i} = \frac{\partial \mathcal{L}_{\text{reg}}(g(\mathbf{z}_i) - y_i)}{\partial \mathbf{z}_i} \tag{28}$$

$$= \frac{\partial \mathcal{L}_{\text{reg}}(g(\mathbf{z}_i) - y_i)}{\partial (g(\mathbf{z}_i) - y_i)} \frac{\partial (g(\mathbf{z}_i) - y_i)}{\partial \mathbf{z}_i} \tag{29}$$

$$= \mathcal{L}_{\text{reg}}'(g(\mathbf{z}_i) - y_i) \nabla g(\mathbf{z}_i). \tag{30}$$

Based on Lemma 1, we have:

$$\nabla g(\mathbf{z}_i)(\mathbf{z}_i' - \mathbf{z}_j') = 0. \tag{31}$$

Thus,

$$\frac{\partial \mathcal{L}_{\text{reg}}}{\partial \mathbf{z}_i}(\mathbf{z}_i' - \mathbf{z}_j') = \mathcal{L}_{\text{reg}}'(g(\mathbf{z}_i) - y_i) \nabla g(\mathbf{z}_i)(\mathbf{z}_i' - \mathbf{z}_j') \tag{32}$$

$$= \mathcal{L}_{\text{reg}}'(g(\mathbf{z}_i) - y_i) \times 0 \tag{33}$$

$$= 0. \tag{34}$$

$\square$

# B DETAILS

## B.1 DETAILS ABOUT THE REAL-WORLD TASKS

For the age estimation on AgeDB-DIR, we adopt the suggested hyper-parameters to train the RankSim, where $\lambda, \gamma$ are set to $2, 1000$, and the results of OE and PH-Reg are adopted from their published papers. The evaluation metric MAE: $\frac{1}{N} \sum_{i=1}^{N} |y_i - y_i'|$, where N is the total number of samples, $y_i, y_i'$ are the label and the predicted result.

For depth estimation on NYUD2-DIR, we adopt the suggested hyper-parameters of OE and PH-Reg to train the models. For RankSim, we train the model with the $\gamma$ range from 1 to 1000. We report the best results for all three baselines. The evaluation metric: threshold accuracy $\delta_1 \triangleq \%$ of $y_p$, s.t. $\max(\frac{y_p}{y_p'}, \frac{y_p'}{y_p}) < 1.25$, and root mean squared error (RMS): $\sqrt{\frac{1}{n} \sum_p (y_p - y_p')^2}$.

# C VISUALIZATIONS

## C.1 VISUALIZATION OF THE UPDATING OF z

We provide the visualization of the feature similarities in Figure. 5.

## C.2 VISUALIZATION OF THE UPDATING OF z

The visualizations of feature manifolds at each epoch are provided in Figure 6. For the neural collapse of regression, the feature manifold will collapse into a single line when the target space is a line and the compression is maximized (Zhang et al., 2024). This trend can be observed in Figure 6, where the feature manifold looks like a thick line and evolves to a thinner line over training. However, standard regression's limited ability to tighten representations results in a slower collapse. In contrast, our proposed method and RankSim both accelerate this collapse, as shown in Figure 3.

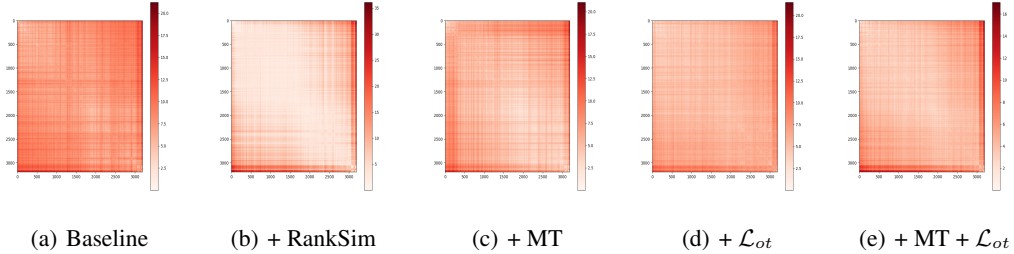

| (a) Baseline | (b) + RankSim | (c) + MT | (d) + $\mathcal{L}_{ot}$ | (e) + MT + $\mathcal{L}_{ot}$ |

Figure 5: Feature similarity matrices (Euclidean Distance). Tightening the representations results in a better ordinality.

## C.3 UPDATING OF MULTIPLE $\theta$

The experiments are conducted on NYUD2-DIR, we change the last hidden layer's feature space to three dimensions for visualization, and the M in our MT strategy is set to 3. The change of multiple $\theta$s throughout the training is shown in Figure 7. We further plot the change of $\{v_{\boldsymbol{\theta}}^i | i = 500, k \in \mathcal{Z}, 0 \le k \le 500\}$ for three $\boldsymbol{\theta}$s. The visualizations are given in Figure 8. The visualization shows the update directions of $\boldsymbol{\theta}$s align with each other, even for a neural network without training.

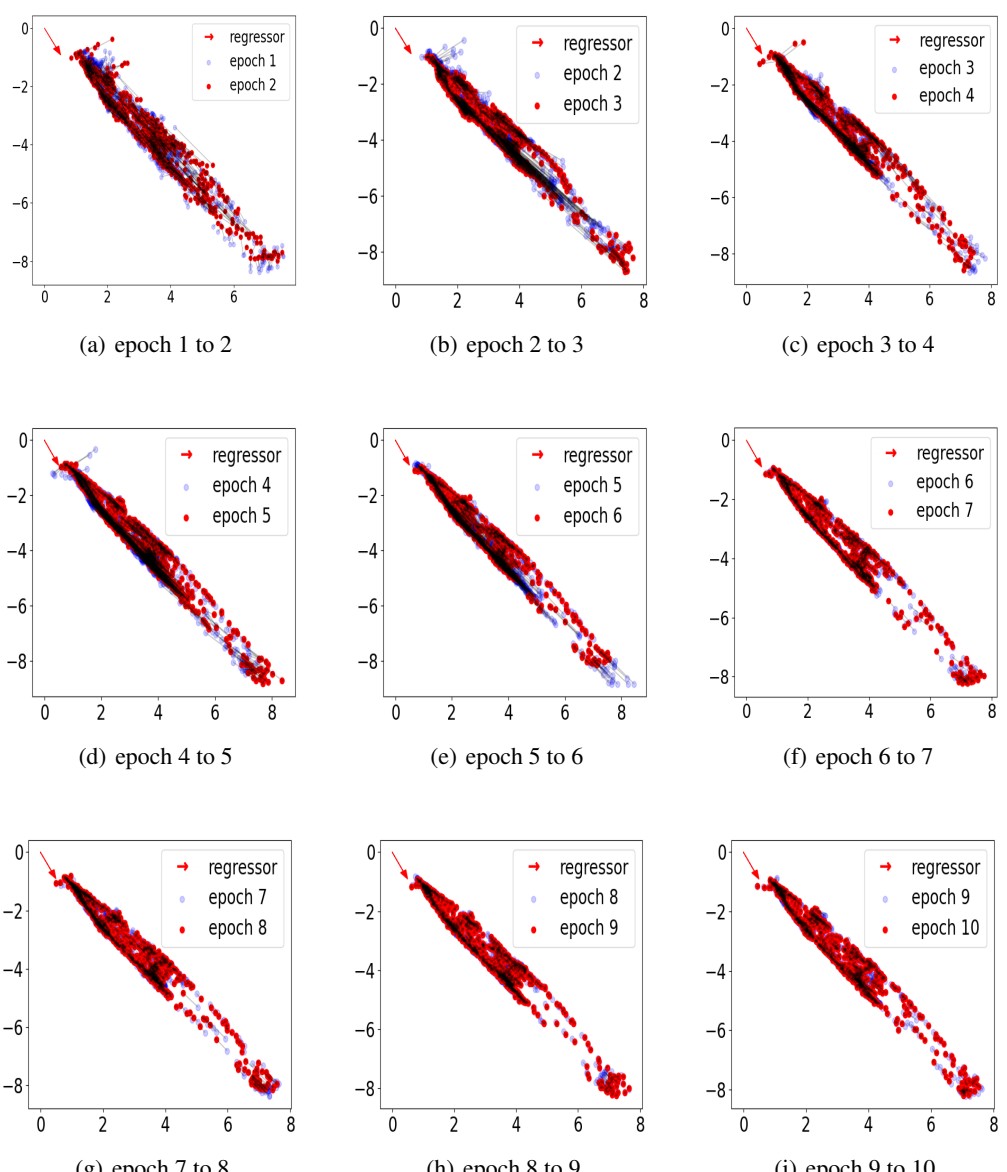

Figure 6: Change of **z** between adjoin epochs.

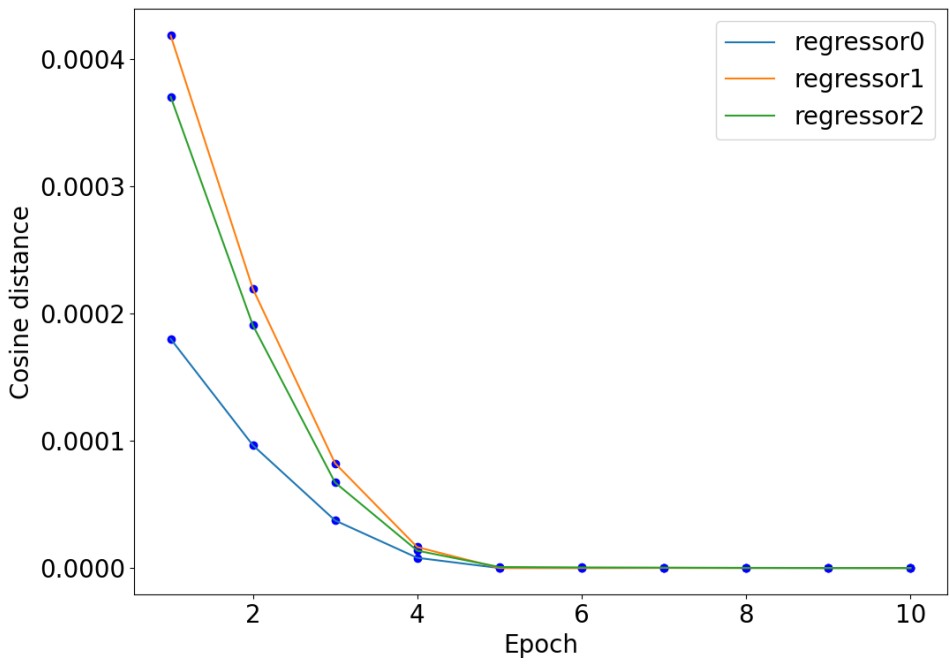

Figure 7: Change of the multiple $\theta$s.

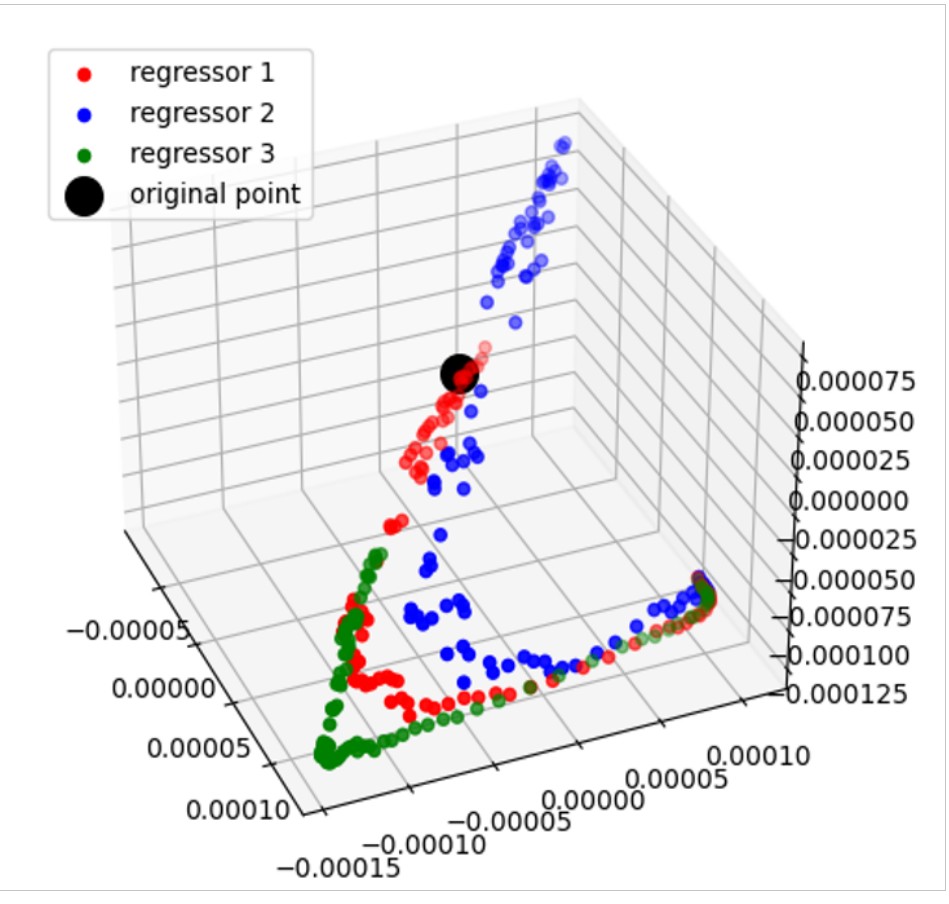

Figure 8: Change of $\theta$s within the iteration $[0, 500]$.

