# OpenReview forum: "Improving Deep Regression with Tightness"
_ICLR.cc/2025/Conference — ICLR 2025 Poster_

### Official Review · Reviewer_CpDH · 2024-10-21

**Soundness:** 3
**Presentation:** 3
**Contribution:** 2
**Rating:** 6
**Confidence:** 4

**Summary:**

This paper proposes a deep regression method focusing on the tightness of features.
The authors found that the tightness of features is important but not sufficiently optimized in normal regression training
because the updated directions of features are limited in normal regression training.
Based on this finding, the multiple target (MT) strategy and regression optimal transport regularizer (ROT-Reg) are proposed to promote tightness during training.
MT increases the number of regression heads so that features are updated toward various directions.
ROT-Reg closes the topology gap between features and targets via self-entropic optimal transport.
Experimental results show that the proposed methods improved the tightness of features and regression performance.

**Strengths:**

- S1: The finding that the updated directions of the features are limited in normal regression training is intriguing.
- S2: The finding is theoretically supported.
- S3: The paper is well-organized and easy to follow.

**Weaknesses:**

- W1: The difference between the global and local tightness is ambiguous. While $\mathcal{H}(\mathbf{Z}|\mathbf{Y})$ is called tightness, formulating the global and local tightness would strengthen the theoretical analysis.
- W2: The justification of design choices of the proposed methods needs to be clarified.
  - For MT, is there a possibility that the multiple regressors will collapse into a single solution? Are the solution spaces $S_y$ orthogonal?
  - For ROT-Reg, is the self-entropic optimal transport the best choice? For example, one may simply minimize the gap of affinity matrices in Eq. (9).
- W3: The performance improvement by the proposed method is marginal. For instance, in Tab. 2, why did incorporating MT and ROT-Reg underperform MT?
- W4: It would be beneficial to visualize the improvement of tightness not only in toy datasets but also in real-world datasets with e.g., PCA.
- W5: Displaying the regression performance in Tab. 4 would be helpful to strengthen the efficacy of tightness on the performance.
- W6: There are many typos. For instance,
  - L185: $\partial$ is missing in partial derivative
  - L113: differentiable entropy -> differential entropy?
  - L252: "$=$" -> "$\in$" ?
  - L317: "performance the performance"

**Questions:**

See the weaknesses.

---

> ### Author Response · Authors · 2024-11-16
> **Thanks for the valuable comments, especially the design choices**
>
> Dear Reviewer CpDH,
>
> Thank you for your insightful and detailed comments. For the weakness:
>
> **W1: Formulating the global and local tightness would strengthen the theoretical analysis.**
>
> Thank you for pointing this out. We have revised the text to clarify this point. Please refer to the revised Section 4 with blue text.
>
> **W2: About the design choices of the proposed methods**
>
> - **For MT, is there a possibility that the multiple regressors will collapse into a single solution? Are the solution spaces orthogonal:**  As discussed in Section 5.4 (Multiple $\theta$s) and Section 3.3, the multiple regressors do not collapse into a single one in reality, as the regressor directions remain almost identical throughout training. This stability arises because the representations are distributed around the regressor weight $\theta$, offsetting each other and thereby having a minimal impact on the direction of the regressor. The solution spaces are not orthogonal to each other, but they are orthogonal to the regressor weight $\theta$ if the regressor is linear.
>
> - **For ROT-Reg, is the self-entropic optimal transport the best choice? For example, one may simply minimize the gap of affinity matrices in Eq. (9).**  The current design is effective but not necessarily the best. Simply minimizing the gap of affinity matrices (i.e., $T^z$ and $T^y$) can introduce optimization challenges, as $T^z$ and $T^y$ are obtained iteratively through the Sinkhorn algorithm rather than simply through gradient backpropagation. In contrast, ROT-Reg is updating $C^z$ through gradient backpropagation to minimize $L_{ot}$. We have revised Section 4.2 to clarify this point (see the corresponding blue text).
>
> **W3. The performance improvement by the proposed method is marginal. For instance, in Tab. 2, why did incorporating MT and ROT-Reg underperform MT?**
>
> Our method achieves higher improvement than various methods on three datasets.  For instance, on AgeDB-DIR, our method outperforms the previous SOTA method, RankSim, with an improvement of +0.52 (ALL) compared to RankSim's +0.18 (ALL).
>
> The lower performance on some metrics comes from the trade-off between the head (with many samples) and tail (with few samples), which is common in imbalanced cases. Thus, although incorporating both MT and ROT-Reg achieves the best performance on RMSE ALL, it shows a lower performance on the head part (Many & Med.), reflecting the trade-off.
>
> **W4: It would be beneficial to visualize the improvement of tightness not only in toy datasets but also in real-world datasets with e.g., PCA.**
>
> The visualization of the tightness in Figure 3 is conducted on the real-world dataset NYUD2-DIR for depth estimation. We have revised our text to state this explicitly.
>
> **W5: Displaying the regression performance in Tab. 4 would be helpful to strengthen the efficacy of tightness on the performance.**
>
> Thanks for pointing this out, we have revised Table 4 to display the regression performance (i.e., RMSE ALL).
>
> **W6: There are many typos.**
>
> Thanks for pointing this out, we have revised some typos including those you mentioned.
>
> We appreciate your valuable comments, especially the design choices,  which have helped us improve the quality and clarity of our work.

---

> > ### Comment · Reviewer_CpDH · 2024-11-21
> >
> > Thank you for your response.
> >
> > For MT, I'm still curious about why multiple $\theta$s do not collapse into a single solution since the feature extractor is also updated.
> > Intuitively, predicting the same target from the same feature seems to make $\theta$s go to the same solution.
> > Describing the reason for `because their directions remain almost identical during training` in L472 would make the paper more convincing.
> >
> > For ROT-Reg, I meant that the affinity matrix is like the cost matrix $\\mathbf{C}$ (or some similarity matrix), e.g., $\\mathcal{L}_{ot}=\\| \\mathbf{C^Z} - \\mathbf{C^Y} \\|_F$.
> > Such a more straightforward loss could encourage consistency.

---

> ### Author Response · Authors · 2024-11-24
> **Thanks for the valuable suggestions**
>
> **Q1. Intuitively, predicting the same target from the same feature seems to make $\theta$s go to the same solution. Describing the reasons why the directions of $\theta$s remain almost identical during training would make the paper more convincing.**
>
> Thank you for your intuition, which aligns with our findings regarding the updating directions of $\theta$s (see discussed in Section 5.4). However, $\theta$s do not collapse into a single solution in practice. This is because **$\theta$s are randomly initialized**, and remain almost identical throughout training (See Figure 4(b) for empirical results), for **three reasons** we outline below. For reference, recall $\frac{\partial L_{reg}}{\partial \theta} =  \frac{1}{b}\sum_{i=1}^b w_iz_i^T$, where $w_i = L'_{reg}(\theta^T z_i - y_i)$:
>
> - **(Update magnitude) The magnitude of $\frac{\partial L_{reg}}{\partial \theta}$ is 'scaled' by $w_i$**, since $w_i$ often follows a Gaussian distribution centered at the origin, as assumed in models like Bayesian linear regression. When $w$ and $z$ are independent or weak dependent, $\mathbb{E}[w_iz_i]$ will approach 0 and causing $\frac{\partial L_{reg}}{\partial \theta}$ be 'scaled' to 0.
>
> - **(Accumulated updates) According to the central limit theorem, the updates of $\theta$ follow a Gaussian distribution.  This causes partial offsets between updates and results in a reduced accumulated effect.** In addition, we empirically observe the mean of this Gaussian distribution approaches 0 (see Figure 4(c)), indicating that $w$ and $z$ are independent or weakly dependent and the **accumulated updates approach 0 (i.e., $\theta$ remain almost identical throughout training)**.
>
> - **(Update Direction) The effect of $z_i$ on the direction of $\frac{\partial L_{reg}}{\partial \theta}$ over a batch of $b$ samples offsets each other** and resulting in the stability of the $\theta$ direction throughout training. This occurs because $z_i$ tends to be distributed around $\theta$, as discussed in the paragraph following Equation 3.
>
> As discussed above, the updates of $\theta$s are limited and thus $\theta$s remain almost identical throughout training.
>
> Thanks for this valuable suggestion. We have revised the text in Section 5.4 to clearly explain these reasons.
>
> **Q2. For ROT-Reg, a more straightforward loss (e.g., $||C^Z - C^Y ||_F$) could encourage consistency.**
>
> Directly minimizing $||C^Z - C^Y ||_F$ imposes an overly strict constraint on the feature manifold, forcing it to become identical to the target space. This is unnecessary, as the feature manifold only needs to be homeomorphic to the target space [1]. We once experimented with this approach, and it did not yield good results. A similar approach has also been implemented by RTD[2], where they minimize the gap between the two matrices on randomly selected edges: https://github.com/danchern97/RTD_AE/blob/714d1b78c6afa976229a08210e3870db4c69f794/src/top_ae.py#L214.
>
> Minimizing the gap between $C^Z$ and $C^Y$ only on paired edges (i.e., edges of the minimal spanning trees) is shown to be more effective, as demonstrated by the topological autoencoder. However, as discussed in Section 4.2, compared to the topological autoencoder, our ROG-Reg captures more local structures of targets when $\gamma>0$. Furthermore, ROG-Reg also performs better than the topological autoencoder, as shown in Table 3.
>
> Thanks for pointing this out, we have revised Section 4.2 to clarify this point.
>
> [1] S. Zhang, K. Kawaguchi, and A. Yao. Deep Regression Representation with Topology. ICML. 2024
>
> [2] I. Trofimov, et al. Learning topology-preserving data representations. ICLR. 2023

---

> > ### Comment · Reviewer_CpDH · 2024-11-26
> >
> > Thanks for your further clarification.
> > My concerns have been addressed.
> > I will update my score to 6.

---

> > > ### Author Response · Authors · 2024-11-26
> > >
> > > Thanks for raising the score. We appreciate your comments and insights about the design choice, which definitely improved the paper.

---

### Official Review · Reviewer_QZxF · 2024-11-01

**Soundness:** 3
**Presentation:** 3
**Contribution:** 3
**Rating:** 6
**Confidence:** 1

**Summary:**

The paper proposes enhancing deep regression by preserving the ordinality of target values within the feature representation space, which is linked to reducing the conditional entropy between the learned representation and the taget. Given typical regressors struggle to naturally reduce the conditional entropy, they propose two solutions for the problem. The first is to use augmenting for the target to mimic training in classifition, so that the model parameters can be with more flexible and diverce updating directions. The second is to use ptimal transport to encourage the similarity between target and feature spaces. Results in some real-world tasks demonstrate the effectiveness of their solutions.

**Strengths:**

1. The paper is overall well-written. The proposal of linking conditional entropy with ordinality preserving  in regression also seems new and interesting.

2. The authors propose the ROT Regularizer and a multi-target learning strategy, both of which are innovative methods for improving regression. These techniques address the limitations of standard regression by refining the feature space structure and better preserving relationships among targets, enabling more robust and accurate predictions.

**Weaknesses:**

I have limited knowledge in this specific area and currently lack the expertise to identify any potential weaknesses in the paper. The overall manuscript appears satisfactory to me. The authors may gain additional insights for refinement through feedback from reviewers more experienced in this topic.

**Questions:**

NA

---

> ### Author Response · Authors · 2024-11-16
> **Thanks for the careful review and constructive suggestions**
>
> Dear Reviewer QZxF,
>
> Thank you for the careful review and constructive suggestions. We are happy that the novelty and contribution of our work are recognized.

---

### Official Review · Reviewer_uwbB · 2024-11-05

**Soundness:** 3
**Presentation:** 3
**Contribution:** 3
**Rating:** 6
**Confidence:** 3

**Summary:**

This work addresses the fundamental yet underexplored problem of regression, arguing that limited research attention has been devoted to it compared to classification tasks. The authors posit that conditional entropy H(Z|Y) , which is extensively studied in information bottleneck, is important for regression problem. They term this entropy measure “tightness” and argue that minimizing it is essential for regression tasks.  Towards tighter representations, the authors propose two strategies, which are multi-target and an optimal transport-based regularize. Experimental results show that both of the proposed strategies can improve the regression performance.

**Strengths:**

+ The proposed method achieves superior performance compared to prior deep regression techniques on two benchmark datasets, demonstrating its effectiveness.

+ The authors provide an interesting analysis on why ordinal feature spaces are not naturally learned under typical regression loss functions, highlighting a crucial aspect often overlooked in regression tasks.

+ The authors offer a comparison between classification and regression, explaining why classification losses tend to better constrain (or “tighten”) representations, which may help readers understand the distinct requirements of these tasks.

**Weaknesses:**

Although the authors discuss why minimizing Mean Squared Error (MSE) may fail to learn ordinal feature spaces, they do not provide empirical results or visualizations, such as t-SNE plots, to support this claim. Comparative visualizations between the proposed method and RankSim would strengthen the discussion

Inconsistency between Eq 3 and Eq 5 – Should both of them be from the batch level? For example, change N to b?

The rationale for employing multiple regressors is unclear. Why is a multi-regressor approach potentially better than a single one? Is this analogous to an ensemble method? Additionally, will the different ranges of Y affect the selection of M?

In [1], the authors discuss the neural collapse phenomenon can be similarly achieved by deep networks when minimizing MSE, which seems somewhat contradictory to statements made in the current paper. Is this contrast due to the nature of the target label (discrete versus continuous)? Expanding on this point with further discussion and comparison would add clarity.

[1] NEURAL COLLAPSE UNDER MSE LOSS: PROXIMITY TO AND DYNAMICS ON THE CENTRAL PATH, ICLR21

**Questions:**

Please see the weakness.

---

> ### Author Response · Authors · 2024-11-16
> **Thanks for the insightful feedback**
>
> Dear Reviewer uwbB,
>
> Thank you for your insightful and detailed comments. For the weakness:
>
> **W1: Although the authors discuss why minimizing Mean Squared Error (MSE) may fail to learn ordinal feature spaces, they do not provide empirical results or visualizations, such as t-SNE plots, to support this claim. Comparative visualizations between the proposed method and RankSim would strengthen the discussion.**
>
> Thanks for pointing this out.  We do provide empirical results and visualizations to support this claim and have now revised our text to state this more explicitly.
> Specifically, Table 4 shows that regression does not effectively preserve ordinality, as indicated by lower Spearman’s and Kendall’s values. However, our proposed method and RankSim are both comparable in preserving ordinality.
> Figure 2 provides visualizations of the feature manifold on the toy dataset for coordinate prediction, showing that our method results in a more tightened and ordinal preserved feature manifold.
> Figure 3 shows similar results on a real-world dataset for depth estimation.
>
>
> **W2. Eq 3 and Eq 5  should both be from the batch level. For example, change N to b.**
>
> Thanks for highlighting this. We have revised the text accordingly.
>
> **W3. Regarding the multiple regressors:**
>
> - **The rationale for employing multiple regressors is unclear:**  The use of multiple regressors is grounded in our theoretical analysis in Section 3.  It is based on our observation that in classification, using multiple classifiers enhances the compression of representations in several directions. Multiple regressors serve to achieve a similar and analogous effect.
>
> - **Why is a multi-regressor approach potentially better than a single one:**
> Similar to the role of multiple classifiers in classification, multiple regressors help tighten the representations in multiple directions, as supported by our theoretical and empirical results.
>
> - **Is this analogous to an ensemble method:**
> No, it is not analogous, although the mean operation may introduce a potential ensemble effect. As discussed in Section 5.5 (mean $\hat{y}$ vs. single $\hat{y}$),
> a single $\hat{y}$ performs nearly the same as ensembling multiple regressors (i.e., mean $\hat{y}$). Therefore, the improvement is not due to the ensemble effect and the method is not analogous to an ensemble method.
>
> - **Will the different ranges of Y affect the selection of M:**  No. The ranges of Y have a limited impact on the selection of M because the primary factor is the intrinsic dimension of the feature manifold, which determines how many additional constraints (i.e., $M-1$) are required to compress the manifold (see Section 4.1 for more details). Our empirical results also indicate this. While the ranges of Y are quite different on NYUD2-DIR (y $\in$ [0.7, 10]) and AgeDB-DIR ( y $\in$ [0, 101]), $M=8$ works well on both datasets.
>
> We have revised our text to state the above points more explicitly. Please refer to the revised Section 5.5 with blue text.
>
> **W4. In [1], the authors discuss the neural collapse phenomenon can be similarly achieved by deep networks when minimizing MSE, which seems somewhat contradictory to statements made in the current paper. Is this contrast due to the nature of the target label (discrete versus continuous)? Expanding on this point with further discussion and comparison would add clarity.**
>
> Our statements do not contradict the neural collapse phenomenon in [1]. Although the model in [1] uses MSE, it is still like a classification model with multiple classifiers/regressors, which allow representation compression in multiple directions.
>
> In cases of neural collapse of regression, the feature manifold will collapse into a single line when the target space is a line and the compression is maximized [2]. This trend can be observed in Figure 6, where the feature manifold looks like a thick line and evolves toward a thinner line over training. However, standard regression’s limited ability to tighten representations results in a slower collapse. In contrast, our proposed method and RankSim both accelerate this collapse, as shown in Figure 3.
>
> We appreciate your insightful feedback, especially the neural collapse, and have included the above discussion in our Section C.2 to improve the clarity and depth of this work.
>
> [1] X.Y. Han, et al. Neural Collapse Under MSE Loss: Proximity to and Dynamics on the Central Path. ICLR. 2022
>
> [2] S. Zhang, K. Kawaguchi, and A. Yao. Deep Regression Representation with Topology. ICML. 2024

---

### Meta-Review · Area_Chair_tDVo · 2024-12-20

**Metareview:**

This work investigates the phenomenon that preserving the ordinality can help regression. According to the success of representation learning for classification, this work finds that tightness of representations corresponds to the ordinality and proposes a regularization to improve the tightness accordingly. While the initial reviews were mixed, the major concerns about ablation experiments, and performance gain were addressed by rebuttal. All reviewers had the positive scores after rebuttal. AC reads the paper, reviews and rebuttal carefully and suggests authors to improve the revision according to suggestions from reviewers.

**Additional Comments On Reviewer Discussion:**

After discussion, the concerns of Reviewer uwbB and CpDH about ablation and performance were addressed and CpDH raised the score to 6. All reviewers leaned to accept the work after discussion.

---

### Decision · Program_Chairs · 2025-01-22

Accept (Poster)